# δ-Opioid Receptor as a Molecular Target for Increasing Cardiac Resistance to Reperfusion in Drug Development

**DOI:** 10.3390/biomedicines11071887

**Published:** 2023-07-03

**Authors:** Natalia V. Naryzhnaya, Alexander V. Mukhomedzyanov, Maria Sirotina, Leonid N. Maslov, Boris K. Kurbatov, Alexander S. Gorbunov, Mikhail Kilin, Artur Kan, Andrey V. Krylatov, Yuri K. Podoksenov, Sergey V. Logvinov

**Affiliations:** 1Cardiology Research Institute, Tomsk National Research Medical Center, Russian Academy of Science, Tomsk 634021, Russiasirotina_maria@mail.ru (M.S.); bobersanker@gmail.com (B.K.K.);; 2Department of Histology, Embryology and Cytology, Siberian State Medical University, Tomsk 634050, Russia

**Keywords:** heart, reperfusion, myocardial infarction, opioid receptors, kinases, K_ATP_ channels, MPT pore

## Abstract

An analysis of published data and the results of our own studies reveal that the activation of a peripheral δ_2_-opioid receptor (δ_2_-OR) increases the cardiac tolerance to reperfusion. It has been found that this δ_2_-OR is localized in cardiomyocytes. Endogenous opioids are not involved in the regulation of cardiac resistance to reperfusion. The infarct-limiting effect of the δ_2_-OR agonist deltorphin II depends on the activation of several protein kinases, including PKCδ, ERK1/2, PI3K, and PKG. Hypothetical end-effectors of the cardioprotective effect of deltorphin II are the sarcolemmal K_ATP_ channels and the MPT pore.

## 1. Introduction

Despite advances in modern cardiology and the use of new diagnostic and treatment technologies, the mortality due to acute myocardial infarctions (AMIs) in cardiology clinics is about 5–7% and has not decreased in recent years [1,2,3]. With the introduction of percutaneous coronary intervention (PCI) in clinical practice, which is used to restore blood flow in the infarct-related coronary artery, reperfusion injury to the heart has become the dominant type of cardiac injury [4]. Highly effective drugs that are approved for clinical use and capable of significantly increasing the heart’s tolerance to reperfusion do not yet exist. The main cause of death in patients with an ST-segment elevation myocardial infarction (STEMI) is cardiogenic shock, for which the mortality rate is 50–70% [5,6]. Therefore, the ideal drug for the prevention of reperfusion injury of the heart should be a pharmacological agent that limits the infarct size and accelerates the recovery of myocardial contractility in reperfusion.

Initially, it was found that δ-OR agonists have a pronounced infarct-limiting effect in vivo [7], a cardioprotective effect in a study with an isolated heart [8], and a cytoprotective effect [9] when opioids were administered before the onset of ischemia. The molecular mechanism of these effects was investigated, which included protein kinases, NO synthase, and K_ATP_ channels. However, it needed to be confirmed that δ-OR agonists could protect the myocardium from reperfusion injury if given after the onset of ischemia.

In 2004, Gross et al. found that the non-selective opioid receptor (OR) agonist morphine (0.3 mg/kg) and the selective non-peptide δ-opioid receptor agonist BW373U86 prevented reperfusion damage to the rat heart in vivo [10]. It has been demonstrated that the infarct-limiting effect of the OR agonist methadone on the reperfusion of rat hearts in vivo was associated with the activation of δ-OR [11]. It should be noted that the maximum dose of morphine permitted for clinical use is 0.1 mg/kg intravenously [12]. It cannot be used at a dose of 0.3 mg/kg due to the risk of side effects. Opioid peptides, unlike morphine, in therapeutic doses do not penetrate the blood–brain barrier (BBB) and, accordingly, do not have central side effects [13]. These facts allowed us to suggest that peptide δ-OR agonists would be more effective than morphine in treatments for patients with AMI and PCI.

The aim of this review was to analyze the published data on the role of δ-OR in the regulation of the cardiac tolerance to reperfusion and the signaling mechanisms of the cardioprotective effect of δ-OR agonists in reperfusion.

## 2. The Role of δ-Opioid Receptors and the Transactivation of EGFR in the Regulation of Cardiac Tolerance to Reperfusion

Gross’s group found that the selective δ-opioid receptor agonist BW373U86 increased the heart’s tolerance to reperfusion, and the infarct-limiting effect of methadone is associated with the activation of δ-OR [7,8]. However, methadone and BW373U86, being small non-peptide molecules, can cross the BBB, so it was not clear whether their infarct-reducing effect was associated with the activation of central or peripheral δ-ORs. Meanwhile, such a possibility could not be ruled out, since morphine has an infarct-limiting effect when it is administered intrathecally [14,15]. δ_1_- and δ_2_-OR agonists exhibit antiarrhythmic properties when administered intracerebroventricularly [16]. The role of ORs on the infarct-sparing effect of BW373U86 in reperfusion is unclear, since the cardioprotective effect of this opioid could be independent of OR stimulation [17]. In addition, it was found that there are two δ-OR subtypes, including δ_1_ and δ_2_ [18]. It has been shown that the stimulation of both δ_1_-OR and δ_2_-OR before coronary artery occlusion increases the cardiac tolerance to ischemia/reperfusion (I/R) [16,17,18]. It should be noted that Gross’s group used the non-peptide δ_1_-OR agonist TAN-67 for the stimulation of δ_1_-OR at a dose of 10 mg/kg intravenously [18,19], which is apparently sufficient to stimulate the central δ_1_-OR. This group did not use naloxone methiodide, an OR blocker that does not cross the BBB, so it remains unclear whether peripheral or central ORs are involved in the infarct-reducing effect of TAN-67 [20,21].

In our previous study, before coronary occlusion, we injected rats with the selective δ_2_-OR peptide agonist deltorphin II and demonstrated that its infarct-limiting effect is associated with the activation of peripheral δ_2_-OR [7]. Rats were subjected to coronary artery occlusion (45 min) and reperfusion (120 min) [22]. The selective peptide δ_2_-OR agonist deltorphin II (0.12 mg/kg) was administered intravenously 5 min before reperfusion. Deltorphin II reduced the infarct size/area-at-risk ratio by about 50%. The non-selective OR antagonist naltrexone abolished the infarct-sparing effect of deltorphin II. Naloxone methiodide, a non-selective OR antagonist that does not penetrate the BBB, eliminated the cardioprotective effect of deltorphin II [22]. Therefore, the infarct-limiting effect of this opioid peptide in reperfusion was associated with the activation of peripheral ORs. The selective δ-opioid receptor antagonist TIPPψ and the selective δ_2_-OR antagonist naltriben completely abolished the infarct-sparing effect of deltorphin II. The selective δ_1_-OR antagonist BNTX, the selective µ-OR antagonist CTAP, and the selective κ-OR antagonist nor-binaltorphimine did not affect the cardioprotective effect of deltorphin II [22]. Therefore, a reduction in the infarct size in reperfusion was achieved due to the activation of peripheral δ_2_-ORs. OR antagonists had no effect on the infarct size. Therefore, endogenous opioid peptides are not involved in the regulation of the cardiac resistance to reperfusion.

The selective δ_1_-opioid receptor agonist DPDPE (at 0.1 and 1 mg/kg) did not affect the infarct size in reperfusion [22]. Since opioid peptides in therapeutic doses do not penetrate the BBB [13], it can be assumed that the activation of peripheral δ_1_-OR does not affect the heart’s tolerance to reperfusion. The non-selective peptide δ-OR agonist DADLE (at 0.088 mg/kg) did not affect the infarct size in reperfusion [22]. However, a study on an isolated rabbit heart demonstrated a cardioprotective effect of the δ-OR agonist DADLE in reperfusion [23]. The δ-opioid receptor agonist BW373U86 at a dose of 0.1 mg/kg did not affect the infarct size, while at a dose of 1 mg/kg, it caused a decrease in the infarct size after the restoration of coronary perfusion [22]. It remains unclear whether the infarction-reducing effect of BW373U86 is associated with the activation of OR and where this receptor is localized in the body. The δ-OR agonist p-Cl-DPDPE had an infarct-limiting effect, but only at a high dose of 1 mg/kg [22]. Consequently, deltorphin II showed the highest activity out of all opioids in reperfusion. It limited the infarct size at a dose of 0.12 mg/kg, while the other opioids reduced the infarct size at a dose of 1 mg/kg.

In a study performed on isolated rat cardiomyocytes subjected to anoxia/reoxygenation, it was found that deltorphin II in a final concentration of 64 nM/l limited the release of the necrosis marker lactate dehydrogenase from cardiomyocytes [22]. Naloxone abolished this cytoprotective effect.

The infarct-limiting effect of deltorphin II was confirmed in rats with diet-induced metabolic syndrome and in aged rats with a coronary artery occlusion (45 min) and reperfusion (120 min) in vivo [24].

Thus, these data convincingly indicate that the activation of peripheral δ_2_-opioid receptors localized in cardiomyocytes increases the resistance of the heart to the pathogenic impact of reperfusion.

Several studies have shown that δ-OR agonists can implement their cardioprotective effect through the transactivation of the epidermal growth factor receptor (EGFR). In a study on an isolated rabbit heart, it was found that the EGFR antagonist AG1478 abolished the cardioprotective effects of the δ-OR agonist DADLE in reperfusion [23].

It has been shown that the EGFR antagonist AG1478 eliminated the infarct-limiting and cytoprotective effects of morphine when it was used in reperfusion [25]. The selective δ-OR antagonist naltrindole reversed the positive effect of morphine in reperfusion. Thus, it is possible that opioids could exert a cardioprotective effect through the transactivation of the δ-OR-EGFR pathway.

## 3. The Signal Mechanism of the Cardioprotective Effect of the Activation of δ-Opioid Receptors

It is known that the activation of protein kinase C (PKC) and phosphatidylinositol-3-kinase (PI3K) plays an important role in the infarct-limiting effect of ischemic and pharmacologic pre- and postconditioning [26,27,28,29]. Mitogen-activated protein kinase kinase 1/2 (MEK1/2), extracellular signal-regulated kinase 1/2 (ERK1/2), protein kinase A (PKA), Janus kinase 2 (Jak-2), and AMP-activated protein kinase (AMPK) are also involved in an increase in the cardiac tolerance to I/R [21,22]. In addition to kinases, NO synthase (NOS) and soluble guanylyl cyclase (GC) are involved in the cardioprotective effect of pre- and postconditioning [21,22]. The hypothetical end-effectors of pre- and postconditioning are the ATP-sensitive K^+^ channels (K_ATP_-channels), the mitochondrial permeability transition pore (MPT pore), and the big conductance Ca^2+^-activated K^+^ channels (BK_Ca_-channels) [21,22].

### 3.1. The Role of Protein Kinases and NO Synthase in the Infarct-Limiting Effect of Opioids

It has been shown that the infarction-limiting and cytoprotective effects of morphine in reperfusion are implemented through the kinase-mechanism phosphorylation of Akt, ERK, and STAT-3 [25]. The selective δ-OR antagonist naltrindole reversed the activation of Akt, ERK, and STAT-3. These data suggest that the activation of δ-OR via transactivation with EGFR leads to the stimulation of the Akt, ERK, and STAT-3 pathways.

In a study on H9c2 cardiomyoblasts, it was found that the δ-OR-dependent phosphorylation of the PI3K-Akt-GSK-3β cascade is mediated via connexin-43, as it did not occur in connexin-43-knockout mice [30].

It was found that AG-490, an inhibitor of Jak-2, eliminates the infarct-reducing effect of morphine and the δ-OR agonist fentanyl isothiocynate, which were administered to rats before a coronary occlusion [31]. It has been reported that the protein kinase A inhibitor H-89 eliminates the infarct-sparing effect of morphine, which was used before a coronary artery occlusion [32]. In 2011, it was demonstrated that morphine used before the ischemia of an isolated heart increased the cardiac resistance to I/R [33]. The AMPK inhibitor compound C eliminated the cardioprotective effect of morphine [33]. Kim et al. demonstrated that the infarct-limiting effect of the κ-OR agonist U50488H in reperfusion depends on ERK1/2 activation [34]. An injection of U50488H prior to cardiac I/R reduced the infarct size, and the PI3K inhibitor wortmannin abolished this effect of opioids [35]. The infarct-limiting effect of the opioid peptide Eribis peptide 94 depends on the activation of inducible NOS [36]. Remifentanil and deltorphin II were administered before coronary occlusion, and it was demonstrated that the infarct-limiting effect of these opioids depends on the activation of PKC [16,29,37]

In most of the listed works, opioids were used before cardiac ischemia, and kinase and NOS inhibitors were administered before the opioids. There are reasons to believe that some of these kinases, enzymes, and molecular structures are involved in the infarct-limiting effect of deltorphin II in reperfusion.

Indeed, it was found that chelerythrine, an inhibitor of most PKC isoforms, abolishes the infarct-limiting effect of deltorphin II during reperfusion [38]. Rottlerin, a selective PKCδ inhibitor, also reversed the cardioprotective effect of deltorphin II (Figure 1). Therefore, PKCδ is involved in the infarct-limiting effect of this peptide. Compound C (an AMPK inhibitor), AG490 (a Jak-2 inhibitor), H-89 (a PKA inhibitor), and L-NAME (an inhibitor of all NOS isoforms) did not affect the infarct-reducing effect of deltorphin II in reperfusion. These data indicate that AMPK, Jak-2, PKA, and NOS are not involved in the deltorphin-induced cardiac tolerance to reperfusion injury (Figure 1). The PKA inhibitor H-89 did not affect the deltorphin II-induced increase in the cardiac tolerance to reperfusion [38]. Therefore, the signaling mechanism of the cardioprotective effect of deltorphin II differs from that of morphine. Wortmannin, a PI3K inhibitor, and PD98059, an inhibitor of the signal cascade consisting of MEK1/2 and ERK1/2, abolished the infarct-sparing effect of the peptide δ_2_-OR agonist [38]. It should be noted that all of the indicated inhibitors did not affect the infarct size in the reperfusion of the heart. Therefore, AMPK, Jak-2, PKA, NOS, MEK1/2, ERK1/2, PKCδ, and PI3K are not involved in the regulation of the cardiac tolerance to reperfusion in non-adapted rats (Figure 1).

### 3.2. The Role of Guanylyl Cyclase in the Cardioprotective Effect of OR Stimulation

It was previously found that the non-selective μ- and δ-OR agonist D-Ala^2^,Leu^5^,Arg^6^-enkephalin (dalargin, 0.1 mg/kg) administered to rats before a coronary artery occlusion increased the cGMP level in the area at risk [39]. Deltorphin II had the same effect on the reperfusion of the heart; under the impact of this peptide, the cGMP level in the area at risk increased by two-fold [38]. The sources of cGMP in the cell are soluble guanylyl cyclase (GC) [40] and the atrial natriuretic peptide (ANP) receptor [41]. The selective μ-OR agonist fentanyl has been reported to increase ANP secretion [40]. There are no such data regarding δ-OR agonists, so it can be assumed that deltorphin II increases soluble GC activity, which is activated by nitric oxide [40]. However, a NOS blockade did not affect the infarct-limiting effect of deltorphin II [38]. The soluble guanylyl cyclase inhibitor ODQ abolished the cardioprotective effect of deltorphin II [38].

Therefore, soluble GC is involved in a δ_2_-OR-mediated increase in the cardiac tolerance to reperfusion. What is the mechanism of the activation of soluble GC in this case? It is possible that the activation of δ_2_-OR promotes the stimulation of heme oxygenase-1, which synthesizes carbon monoxide, and CO activates soluble GC [7,36]. The protective role of HO-1 in an I/R injury of the heart has been demonstrated [42]. Castany et al. found that the δ_1_-OR agonist DPDPE, when it was administered subcutaneously to mice at a dose of 5 mg/kg, had an antinociceptive effect that was abolished by protoporphyrin IX tin, a heme oxygenase-1 inhibitor [43]. The researchers used a large dose of DPDPE (5 mg/kg), so it is possible that, at this dose, the peptide activated not only the δ_1_-OR, but also the δ_2_-OR or that it passed through the BBB and stimulated the central OR. It has been reported that a morphine-induced decrease in the intraocular pressure is associated with the activation of heme oxygenase-1 [44]. Therefore, the hypothesis about the involvement of heme oxygenase-1 and soluble GC in the infarct-limiting effect of deltorphin II is justified. It should be noted that it is still unknown how the signal from OR is transmitted to heme oxygenase-1 (Figure 1).

### 3.3. The Role of Reactive Oxygen Species in the Cardioprotective Effect of Opioids

Reactive oxygen species (ROS) can damage cardiomyocytes [45] and mediate the arrhythmogenic impact of I/R [46,47]. However, ROS can also protect the heart against I/R due to the activation of PKC and other kinases [48]. Preconditioning, physical exercises [49], and several metabolites, such as bradykinin, acetylcholine, phenylephrine, and opioids, can trigger myocardial protection via an ROS-dependent mechanism [50].

However, researchers have previously associated the protective effect of δ-OR agonists with a decrease in ROS production. It was shown that the anti-apoptotic effect of the selective δ-OR agonist D-Ala^2^,D-Leu^5^-enkephalin (DADLE) during the glucose deprivation of primary cultures of neonatal cardiac myocytes was accompanied by a decrease in ROS production [51]. In addition, the prior administration of the δ-OR agonist resulted in a decrease in the diene conjugate levels and a decrease in superoxide dismutase (SOD) activity in the isolated rat heart after oxidative stress induction by Fe^2+^-ascorbate [52].

However, it was found that ROS formation under the impact of opioids could be responsible for protective redox signaling. Thus, in 2007, it was found that a pretreatment with the selective agonist δ-OR DADLE was able to limit the infarct size, increase ROS production in the myocardium in I/R, and lead to an increase in the phosphorylation of PI3K and AKT [53]. It should be noted that studies with antioxidants or free radical scavengers were not performed in this investigation.

The δ-opioid receptor agonist DADLE protected H9c2 cells against the H_2_O_2_- and antimycin-A-induced injury of cells [30]. The stimulation of protective redox signaling was accompanied by an increase in mitochondrial ROS formation. At the same time, investigators did not carry out a study with antioxidants or ROS scavengers, so the significance of ROS in the cytoprotective effect of DADLE was not proven.

In a study by Xu in 2022, in a model of global I/R in isolated rat hearts, a cardioprotective effect of morphine was found when it was administered after ischemia in the first 30 min of reperfusion and in the reoxygenation of isolated cardiomyocytes [25]. At the same time, morphine caused an increase in the ROS level and superoxide (test with MitoSOX Red). The use of the selective δ-OR antagonist naltrindole inhibited the morphine-induced activation of AKT, ERK1/2, and STAT-3. These data show that δ-OR is involved in the protective effect of morphine. However, this study does not provide evidence that naltrindole prevents ROS formation, which occurs under the impact of morphine. Antioxidants and free radical scavengers were not used in this study. Thus, the authors were able to show that morphine causes an increase in the ROS generation (redox signaling), but the significance of this signaling on the protective effect of morphine and the role of δ-OR in this case have not been proven. It was shown that morphine is able to activate the (PI3K)/Akt signaling pathway in H9c2 cells, which was prevented by the use of the free radical scavenger 2-mercaptopropionyl glycine (2-MPG) [54]. Thus, only indirect evidence confirms the role of ROS in the protective effect of δ-OR agonists in reperfusion. Tsutsumi et al. found that the free radical scavenger 2-MPG abolished the infarct-limiting effect of the δ-OR agonist SNC-121 [55]. However, these investigators did not carry out a receptor analysis of the effect of SNC-121 on ROS formation. At the same time, it was reported that SNC-121 could have a cardioprotective effect that is not associated with opioid receptors [56]. Therefore, the relationship among the activation of δ-OR in reperfusion, ROS formation, and cardioprotection remains a hypothesis.

We found that the cardioprotective effect of deltorphin II persisted after a 2-MPG injection [38]. Therefore, free radicals are not involved in the infarct-reducing effect of deltorphin II. It can be assumed that the signaling mechanisms of the protective effect of deltorphin II and SNC-121 are different. Some researchers suggest that cardioprotection during a long-term adaptation to hypoxia is implemented through the ability of endogenous δ-opioid receptor agonists to increase ROS production. Thus, it was found that naltrindole and the antioxidant N-acetylcysteine prevented cardioprotection in chronic hypoxia [57].

These data show that the widely accepted opinion about the important role of ROS in cardioprotection under the impact of δ-OR agonists has not yet been confirmed when these agonists were used after the onset of ischemia before or during reperfusion. The role of redox signaling in the cardioprotective effect of δ-OR agonists in reperfusion requires an additional study. Perhaps the use of more effective methods for measuring ROS [47,58] will contribute to achievement in this issue.

### 3.4. Hypothetical End-Effectors of the Cardioprotective Action of Opioids and Their Effect on Cell Death Pathways

We have already reported above that hypothetical end-effectors can be K_ATP_ channels, the MPT pore, or BK_Ca_ channels [21,22]. What is the role of these molecular structures in the infarct-limiting effect of opioids?

Mitochondrial K_ATP_ channels (mitoK_ATP_ channels) are reported to be involved in the infarct-reducing effect of the κ-OR agonist U50488H administered prior to ischemia [59]. MitoK_ATP_ channels are involved in the infarct-limiting effect of deltorphin II, which is administered before a coronary occlusion [7]. Sarcolemmal K_ATP_ channels (sarcK_ATP_ channels) and mitoK_ATP_ channels are involved in the cardioprotective effect of the EP 94 opioid [36]. A nonselective antagonist of sarcK_ATP_ channels and mitoK_ATP_ channels, glibenclamide, eliminated the cardioprotection induced by the δ-OR agonist BW373U86 [60]. There is evidence that κ-OR activation increases the cardiac tolerance to reperfusion due to mitoK_ATP_ channel opening [61]. In general, it is widely accepted that the infarct-sparing effect of opioids in I/R is associated with mitoK_ATP_ opening; the prevention of a fall in mitochondrial Δψ in I/R [62,63]; a reduction in the Ca^2+^ overload of cardiomyocytes [64]; the improvement in the cardiac contractility at reperfusion [65]; and a decrease in the necrosis and apoptosis of cardiomyocytes [66]. Apoptosis plays an important role in reperfusion injury of the heart [67,68,69].

However, in our previous study, it was found that the mitoK_ATP_ channel blocker 5-hydroxydecanoate did not affect the infarct-sparing effect of deltorphin II in reperfusion [38]. The sarcK_ATP_ channel inhibitor HMR1098 completely abolished the cardioprotective effect of deltorphin II. Therefore, deltorphin-induced cardiac resistance to reperfusion is associated with sarcK_ATP_ channel opening. It was reported that sarcK_ATP_ opening can prevent the apoptosis and necrosis of cardiomyocytes in ischemia-reperfusion of the heart [70]. SarcK_ATP_ opening was associated with the prevention of cardiomyocyte Ca^2+^ overload through the hyperpolarization of the cell membrane [71].

The MPT pore is an important regulator of cardiac tolerance to I/R [21,22], including in diabetic hearts [72]. MPT pore opening enhances the myocardial damage in I/R, and a blockage of the MPT pore increases the heart’s tolerance to I/R [4]. It is believed that MPT pore opening occurs in reperfusion, which leads to reperfusion damage to the heart [4]. In a study performed on isolated rat hearts, it was demonstrated that the infarct-limiting effect of morphine in reperfusion is a consequence of MPT pore closing [73]. Other investigators obtained similar data for isolated rat hearts; they found that δ_1_-OR and the MPT pore are involved in the cardioprotective effect of morphine [74]. The MPT pore blocker atractyloside also reversed the infarct-limiting effect of deltorphin II in reperfusion [38]. The closed state of the MPT pore promotes Ca^2+^ retention with mitochondria. In ischemia-reperfusion, the transport and accumulation of Ca^2+^ by mitochondria can compensate for a calcium overload in the sarcoplasm and reduce I/R injury [75,76]. The closed state of the MPT pore prevents a fall in Δψ, promotes ATP synthesis, and also prevents the release of proapoptotic factors [77]. In addition, the MPT pore acts as a regulator of mitochondrial dynamics [78]. These mechanisms could be involved in the protection of the heart in reperfusion.

The role of BK_Ca_ channels in the infarct-limiting effect of opioids has not been studied until recently. Paxilline, a BK_Ca_ channel blocker, did not affect deltorphin-induced cardiac tolerance to reperfusion [38].

Therefore, δ-opioid-induced cardiac resistance to reperfusion is associated with sarcK_ATP_ channel opening and MPT pore closing (Figure 1).

Most of the aforementioned studies evaluated the protective effect of δ-OR agonists by assessing the infarct size with tetrazolium or by using the release of intracellular enzymes into the perfusion solution of an isolated heart or in the incubation medium of cardiomyocytes, which are used as a manifestation of cell necrosis. Thus, the important mechanism of cardioprotection by δ-OR agonists is the prevention of cardiomyocyte necrosis. However, apoptosis is an important pathway of cell death in reperfusion. The effect of δ-OR agonists on the state of the MPT pore suggests that they could abolish apoptosis. Indeed, a few studies have shown that δ-OR agonists can reduce cardiomyocyte apoptosis in ischemia-reperfusion [51,79]. However, these studies used a pretreatment with δ-OR agonists. Thus, the effect of δ-OR agonists on the apoptosis of cardiomyocytes in reperfusion was not proven.

It was found that δ-OR agonists can affect autophagy. BW373U86 upregulated autophagy and protected cardiomyocytes against H/R injury, which could have been related to the stimulation of the PI3K/Akt/mTOR pathway [80].

Unfortunately, there are no data on the effect of δ-OR agonists on other cell death pathways.

## 4. Conclusions

These data indicate that the stimulation of a peripheral δ_2_-opioid receptor localized in cardiomyocytes increases the heart’s tolerance to reperfusion. The cardioprotective signal is transmitted from δ_2_-OR to intracellular structures (sarcK_ATP_ channels and the MPT pore). This effect could be mediated by PKSδ, ERK1/2, PI3K, soluble GC, and cGMP-sensitive protein kinase G (PKG). It is possible that these kinases phosphorylate the sarcK_ATP_ channels and the MPT pore, which leads to a change in their properties. It should be noted that this hypothesis needs experimental evidence. Many questions still remain unanswered. It is not clear whether deltorphin II actually activates heme oxygenase-1. If so, what is the signaling mechanism in this case? How is the signal transmitted from δ_2_-OR to PKCδ, ERK1/2, PI3K, and PKG? One of the most unclear issues is the involvement of ROS and redox signaling in the cardioprotective effect of δ-OR agonists during reperfusion. Thus, the widely accepted opinion about the important role of ROS in the protective effect of δ-OR agonists in reperfusion has not been proven and requires further study. Infarct size measurements in ischemia/reperfusion of the heart are based on the determination of the volume of the myocardium containing cells with a ruptured membrane. However, the rupture of cell membranes occurs not only in necrosis, but also in necroptosis, pyroptosis, and ferroptosis. Whether deltorphin II can inhibit these processes is unknown.

In summary, the currently available evidence suggests that δ-OR is a perspective molecular target for the generation of drugs that limit reperfusion injury of the heart. Meanwhile, it should be noted that the number of selective δ-OR agonists suitable for in vivo use is small. At the same time, it is necessary to create a larger number of new peptide δ-OR agonists that are resistant to enzymatic degradation.

## Figures and Tables

**Figure 1 biomedicines-11-01887-f001:**
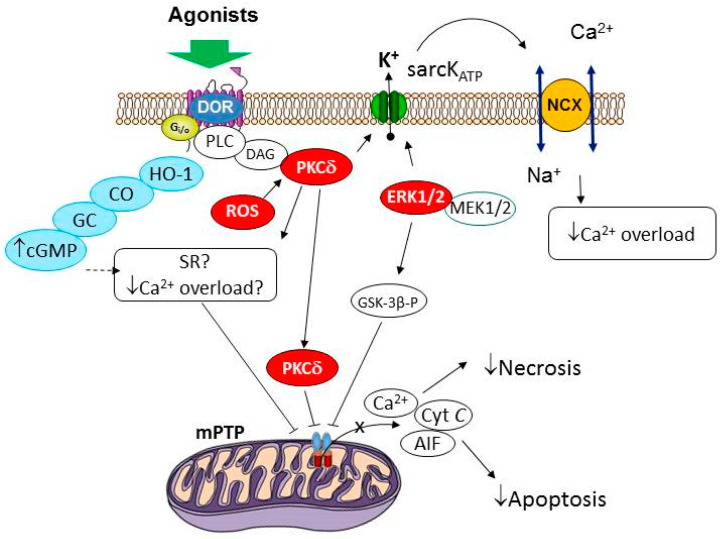
The signal mechanism of the cardioprotective effect of the activation of δ-opioid receptors at reperfusion. Notes: DAG, diacylglycerol; DOR, delta opioid receptor; ERK1/2, extracellular signal-regulated kinase 1/2; MEK1/2, mitogen-activated protein kinase kinase; HO-1, heme oxygenase-1; CO, carbon monoxide; GC, soluble guanylyl cyclase; Gi/o, G-protein; GSK-3β-p, phosphorylated glycogen synthase kinase 3β; sarcK_ATP_, sarcolemmal ATP-sensitive potassium channel; mPTP, mitochondrial permeability transition pore; NCX, sodium calcium exchanger; PKCδ, protein kinase C δ; ROS, reactive oxygen species; PLC, phospholipase C; SR, sarcoplasmic reticulum.

## Data Availability

The data presented in this study are available on request from the corresponding author.

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
