# Peer review of "δ-Opioid Receptor as a Molecular Target for Increasing Cardiac Resistance to Reperfusion in Drug Development"

_biomedicines, 2023, doi:10.3390/biomedicines11071887_

Round 1

Reviewer 1 Report

The manuscript is a concise review positing the DOR as a molecular target to limit reperfusion injury.

With discussion focusing on signalling and end-effectors, a signalling schematic would be beneficial.

The section on hypothetical end-effectors needs some more detail on 'how'. For example, much of the section is about mitoKATP, yet no information is provided about how mitoKATP opening might elicit cardioprotection.

Moreover, the Patel paper (FASEB J  2002 Sep;16(11):1468-70) would be a beneficial addition to bring the focus back to the DOR.

The conclusion would benefit from addressing the title - that is, the DOR as a molecular target for generation of drugs to limit reperfusion injury.

Intro: the sentence starting with 'introduction of percutaneous..." requires revision. Perhaps as simple as beginning the sentence with 'With the introduction....'?

line 63: "In our study..." would perhaps be better as 'In our previous study....'?

line 79: it is unclear what is meant by 'non-adapted rats'.

line 97: 'old rats' would be better reported as 'aged rats'

Author Response

Dear reviewer!

The authors thank you for evaluating our article and for comments. We tried to correct the manuscript as much as possible according with your comments.

Your remarks:

The manuscript is a concise review positing the DOR as a molecular target to limit reperfusion injury.

With discussion focusing on signalling and end-effectors, a signalling schematic would be beneficial.

  • We added scheme.

The section on hypothetical end-effectors needs some more detail on 'how'. For example, much of the section is about mitoKATP, yet no information is provided about how mitoKATP opening might elicit cardioprotection.

  • We have added a brief discussion of end effector cardioprotection options.Unfortunately, no role for mitoKATP was found in our previous data (doi:10.3390/membranes13010063.) On this reason we discussed only sarcKATP and mPTP mechanisms.

Moreover, the Patel paper (FASEB J  2002 Sep;16(11):1468-70) would be a beneficial addition to bring the focus back to the DOR.

  • We have cited this paper.

The conclusion would benefit from addressing the title - that is, the DOR as a molecular target for generation of drugs to limit reperfusion injury.

  • We have added this

Intro: the sentence starting with 'introduction of percutaneous..." requires revision. Perhaps as simple as beginning the sentence with 'With the introduction....'?

  • We have changed this sentences.

line 63: "In our study..." would perhaps be better as 'In our previous study....'?

  • We have changed this sentence.

line 79: it is unclear what is meant by 'non-adapted rats'.

  • We have removed those words.

line 97: 'old rats' would be better reported as 'aged rats'

  • We have changed these words.

Reviewer 2 Report

In this interesting review, the Authors highlight the role played by δ-opioid Receptor in reducing IR damage. The report is interesting and well structured.

Points

1- lacks the link between opiods/redox signaling which, as is known, plays an important role in cardioprotection. Authors can mention in the discussion appropriately this aspect (See and quote please (PMID: 19248760; PMID: 31892282)

2- a link between mitochondrial ATP-sensitive K(+) channel/δ-opioid receptor is present and reported in the literature; it would therefore be appropriate for the authors to discuss this aspect given the importance of these channels in reducing IR damage

3- insert a table highlighting the δ-opioid Receptor role in the three different cardioprotection protocols (pre, post and remote)

4 insert a figure in which the molecular pathway involved is highlighted.

Author Response

Dear reviewer!

The authors thank you for evaluating our article and for comments. We tried to correct the manuscript as much as possible according with your comments.

Your remarks:

1- lacks the link between opiods/redox signaling which, as is known, plays an important role in cardioprotection. Authors can mention in the discussion appropriately this aspect (See and quote please (PMID: 19248760; PMID: 31892282)

  • We have added recommended citation and a short explanation. However, the role of redox signaling in deltorphin-2 cardioprotection at reperfusion has not yet been fully established. Thus, we did not obtain the abolition of the infarct-limiting effect of deltorphin-2 during reperfusion against the background of 2-mercaptopropionyl glycine (doi:10.3390/membranes13010063). I think the role of redox signaling requires additional research, which we noted in the article.

2- a link between mitochondrial ATP-sensitive K(+) channel/δ-opioid receptor is present and reported in the literature; it would therefore be appropriate for the authors to discuss this aspect given the importance of these channels in reducing IR damage

  • We added short discussion of this point.

3- insert a table highlighting the δ-opioid Receptor role in the three different cardioprotection protocols (pre, post and remote)

  • Since the article is devoted to the cardioprotection of DOR agonists, we do not cover their role in cardioprotection during pre-, post-, and remote conditioning. This has been the subject of our previous articles (DOI 10.1007/s11055-016-0236-7, 10.1007/s11055-017-0453-8; doi: 10.2174/1573403X15666190226095820.

4 insert a figure in which the molecular pathway involved is highlighted.

  • We have added scheme with δ-opioid receptor pathway

Reviewer 3 Report

Naryzhnaya NV et al., Biomedicines, Review

The authors, in their short review, analyzed the role of opioids, focusing on δ-OR, in the regulation of cardiac tolerance to reperfusion-induced injury (infarct size) and analyzing the signaling mechanisms (e.g., reactive oxygen species, NOS, KATP channels, PKC) of δ-OR agonists in the myocardium. The authors concluded that the size of the necrosis (infarct size) during ischemia/reperfusion is based on the determination of the mass of myocardium containing cells and membranes ruptured. It is also emphasized that rupture of cell membranes occurs not only during necrosis, but also in various “ptosis”, including e.g., pyroptosis, necroptosis, and ferroptosis. How deltorphin II may inhibit these processes, separately or together, are unknown.

The review manuscript is well written and having several subchapters. It must be emphasized that the estimation and checking of grammatical errors and typos were not the duty of the reviewer. Only, the scientific value of the manuscript is evaluated by this reviewer.

The analysis and the signal mechanisms of the review is not very new, however, several and valuable data are included and summarized in this manuscript.

Various and several drugs can modify the reperfusion-induced injury. At least ONE FIGURE must be incorporated, including some appropriate references (please, see below) in the revised version, e.g., what the consequences are involved in the reperfusion-induced injury. Such as cell deaths; necrosis, apoptosis, autophagy, pyroptosis, ferroptosis and cardiac arrhythmias. 

Page 3, from line 102: Subchapter Number 3 (“The signal mechanism of the cardioprotective effect …..”) is too short. This subchapter should be extended.

Page 4, from line 173: Subchapter Number 6 (“The role of reactive species in …..”) is also too short. It must be extended in the revised version of this manuscript.

Additional suggestions:

It has been very well known for several decades that many drugs and interventions (e.g., opioids, NO, KATP channels, reactive oxygen species, heme oxygenase, CO, preconditioning) could modify the consequences and final outcomes of reperfusion-induced injury. Reactive oxygen species (ROS) and other drugs play also a critical role and promote signal transduction (please, see the references below) in reperfusion-induced injury, including infarct size limitation and arrhythmogenesis (J Pharmacol Exp Ther, 1986 Apr;237(1):1-8, PMID: 3958960; Eur J Pharmacol. 1987 Nov 17;143(3):391-401. doi: 10.1016/0014-2999(87)90463-8; Br J Pharmacol. 1995 Nov;116(6):2597-602. doi: 10.1111/j.1476-5381.1995.tb17213.x; Eur J Pharmacol. 1995 Jun 12;279(2-3):251-6. doi: 10.1016/0014-2999(95)00164-g; Walsh SK, Br J Pharmacol. 2010 Jul;160(5):1234-42. doi: 10.1111/j.1476-5381.2010.00755.x; Juhasz B, Varga B, J Cell Mol Med. 2011 Sep;15(9):1973-82. doi: 10.1111/j.1582-4934.2010.01153.x; Ter Horst EN, Lab Anim. 2018 Jun;52(3):271-279. doi: 10.1177/0023677217724485. Epub 2017 Aug 4; Front Pharmacol. 2020 May 12;11:616. doi: 10.3389/fphar.2020.00616. eCollection 2020; Front Physiol. 2022 Mar 14;13:856699. doi: 10.3389/fphys.2022.856699. eCollection 2022).

THE DIRECT measurements of ROS were initially carried out by several research teams in the past, using ESR (electron spin resonance) and EPR (electron paramagnetic resonance) spectroscopies in animal models in the myocardium. Independent of the experimental model used, the consensus has been reached that ROS generation/or oxidative stress is the most important signal mechanism in reperfusion-induced injury.

Proc Natl Acad Sci U S A. 1987 Mar;84(5):1404-7. doi: 10.1073/pnas.84.5.1404;

Circ Res,. 1987 Nov;61(5):757-60.  doi: 10.1161/01.res.61.5.757

Free Radic Biol Med. 1990;8(4):363-72. doi: 10.1016/0891-5849(90)90102-o

J Magn Reson. 2021 Aug;329:107024. doi: 10.1016/j.jmr.2021.107024. Epub 2021 Jun 9

How it is possible, as the authors by Naryzhnaya NV et al., et al., described that “Therefore, free radicals are not involved in the infarct-reducing effect of deltorphin II.” (page 4, lines 178 to 179) in the reperfusion-induced injury. ROS have the most harmful effects on cell membrane and membrane integrity. This discrepancy showed be explained (page 4, lines 178 to 179) in detail and discussed/acknowledged the aforementioned publications in the revised version of the manuscript.

Necrosis-, apoptosis- and autophagy-induced cell deaths and their mechanisms are the major processes in the malfunction of myocardial tissues induced by oxidative stress in reperfusion. Thus, the following “classic and recent” publications (please, see below) also should be cited and acknowledged, which may substantially increase the interest of general readers, including the clinicians and experimental researchers:

-      Circulation. 1998 Jan 27;97(3):276-81. doi: 10.1161/01.cir.97.3.276

-      Apoptosis. 2006 Dec;11(12):2195-204. doi: 10.1007/s10495-006-0292-5

-      J Cell Mol Med. 2013 Aug;17(8):936-57. doi: 10.1111/jcmm.12074. Epub 2013 Jun 22

-      Front Pharmacol. 2022 Nov 14;13:964475. doi: 10.3389/fphar.2022.964475. eCollection 2022

-      Free Radic Biol Med. 2023 Feb 1;195:219-230. doi: 10.1016/j.freeradbiomed.2022.12.097. Epub 2022 Dec 30

The incorporation of the aforementioned publications and the suggested Figure in the revised version of the manuscript may substantially increase the interest of general readers, including the clinicians and experimental researchers.

The review manuscript is well written and having several subchapters. It must be emphasized that the estimation and checking of grammatical errors and typos were not the duty of the reviewer. Only, the scientific value of the manuscript is evaluated by this reviewer.

Author Response

Dear reviewer!

The authors thank you for evaluating our article and for comments. We tried to correct the manuscript as much as possible according with your comments.

At least ONE FIGURE must be incorporated, including some appropriate references (please, see below) in the revised version, e.g., what the consequences are involved in the reperfusion-induced injury. Such as cell deaths; necrosis, apoptosis, autophagy, pyroptosis, ferroptosis and cardiac arrhythmias. 

  • We have added a scheme of the mechanism of the cardioprotective action of DOR agonists.

Page 3, from line 102: Subchapter Number 3 (“The signal mechanism of the cardioprotective effect …..”) is too short. This subchapter should be extended.

  • Sorry, the article was initially incorrectly numbered sections.Thus it turned out that the section includes only 9 lines.This, of course, is extremely small for a large amount of material.We have changed the numbering of the section and thus the section includes more than 40 literary sources. In addition, this section has been supplemented by the links you suggested.

Page 4, from line 173: Subchapter Number 6 (“The role of reactive species in …..”) is also too short. It must be extended in the revised version of this manuscript.

  • We have added a small addition to this chapter. However, the role of redox signaling in deltorphin-2 cardioprotection at reperfusion has not yet been fully established. Thus, we did not obtain the abolition of the infarct-limiting effect of deltorphin-2 during reperfusion against the background of 2-mercaptopropionyl glycine (doi:10.3390/membranes13010063). I think the role of redox signaling requires additional research, which we noted in the article.

Additional suggestions:

It has been very well known for several decades that many drugs and interventions (e.g., opioids, NO, KATP channels, reactive oxygen species, heme oxygenase, CO, preconditioning) could modify the consequences and final outcomes of reperfusion-induced injury. Reactive oxygen species (ROS) and other drugs play also a critical role and promote signal transduction (please, see the references below) in reperfusion-induced injury, including infarct size limitation and arrhythmogenesis

  • We have cited the following publications, we thank the reviewer, and they were very useful for revealing the topic of the review:

Juhasz B, Varga B, J Cell Mol Med. 2011 Sep;15(9):1973-82. doi: 10.1111/j.1582-4934.2010.01153.x;

Front Pharmacol. 2020 May 12;11:616. doi: 10.3389/fphar.2020.00616. eCollection 2020;

Front Physiol. 2022 Mar 14;13:856699. doi: 10.3389/fphys.2022.856699. eCollection 2022.

However following paper were not cited by us, sorry.

  • J Pharmacol Exp Ther, 1986 Apr;237(1):1-8, PMID: 3958960;

Unfortunately, this paper is not related to the topic of our review.

  • Eur J Pharmacol. 1987 Nov 17;143(3):391-401. doi: 10.1016/0014-2999(87)90463-8;

Unfortunately, this paper is not related to the topic of our review.

  • Br J Pharmacol. 1995 Nov;116(6):2597-602. doi: 10.1111/j.1476-5381.1995.tb17213.x;

Unfortunately, this paper is not related to the topic of our review.

  • Eur J Pharmacol. 1995 Jun 12;279(2-3):251-6. doi: 10.1016/0014-2999(95)00164-g;

Unfortunately, this paper is not related to the topic of our review.

  • Walsh SK, Br J Pharmacol. 2010 Jul;160(5):1234-42. doi: 10.1111/j.1476-5381.2010.00755.x;

In this study, cannabinoid receptor agonist was administered prior to the onset of ischemia. At the same time, our article is devoted to the effect of delta agonists when administered after ischemia to verify their effect on reperfusion injury. Thus, we can offer delta-OR agonists for the treatment of acute myocardial infarction, when the patient cannot be given a drug before ischemia, but has to deal with already formed ischemic damage. At the same time, it is still possible to influence the reperfusion injury; for this, we propose to use DOR agonists.

  • Ter Horst EN, Lab Anim. 2018 Jun;52(3):271-279. doi: 10.1177/0023677217724485. Epub 2017 Aug 4;

In this study, mu-OR agonists were administered prior to the onset of ischemia. At the same time, our article is devoted to the effect of delta agonists when administered after ischemia to verify their effect on reperfusion injury. Thus, we can offer delta-OR agonists for the treatment of acute myocardial infarction, when the patient cannot be given a drug before ischemia, but has to deal with already formed ischemic damage. At the same time, it is still possible to influence the reperfusion injury; for this, we propose to use DOR agonists.

THE DIRECT measurements of ROS were initially carried out by several research teams in the past, using ESR (electron spin resonance) and EPR (electron paramagnetic resonance) spectroscopies in animal models in the myocardium. Independent of the experimental model used, the consensus has been reached that ROS generation/or oxidative stress is the most important signal mechanism in reperfusion-induced injury.

Proc Natl Acad Sci U S A. 1987 Mar;84(5):1404-7. doi: 10.1073/pnas.84.5.1404;

Circ Res,. 1987 Nov;61(5):757-60.  doi: 10.1161/01.res.61.5.757

Free Radic Biol Med. 1990;8(4):363-72. doi: 10.1016/0891-5849(90)90102-o

J Magn Reson. 2021 Aug;329:107024. doi: 10.1016/j.jmr.2021.107024. Epub 2021 Jun 9

  • Many thanks for the interesting links!

Unfortunately we do not have such equipment.We made attempts to study ROS by fluorescent methods in myocardial homogenates (2,3-dihydrodichlorofluorescein and Amplex Red).However, it should be noted that these methods have low selectivity in the tissue homogenate and cannot be used to detect fast effects such as redox signaling.In the future, we plan to study reactive oxygen species in a model of hypoxia-reoxygenation of isolated adult rat cardiomyocytes with the addition of the DOR agonist deltorphin-II during reperfusion.In a similar study, we revealed the cytoprotective effect of deltorphin-2.

How it is possible, as the authors by Naryzhnaya NV et al., et al., described that “Therefore, free radicals are not involved in the infarct-reducing effect of deltorphin II.” (page 4, lines 178 to 179) in the reperfusion-induced injury. ROS have the most harmful effects on cell membrane and membrane integrity. This discrepancy showed be explained (page 4, lines 178 to 179) in detail and discussed/acknowledged the aforementioned publications in the revised version of the manuscript.

  • I'll repeat my words above: the role of redox signaling in deltorphin-2 cardioprotection at reperfusion has not yet been fully established. Thus, we did not obtain the abolition of the infarct-limiting effect of deltorphin-2 during reperfusion against the background of 2-mercaptopropionyl glycine (doi:10.3390/membranes13010063). I think the role of redox signaling requires additional research, which we noted in the article.

Necrosis-, apoptosis- and autophagy-induced cell deaths and their mechanisms are the major processes in the malfunction of myocardial tissues induced by oxidative stress in reperfusion. Thus, the following “classic and recent” publications (please, see below) also should be cited and acknowledged, which may substantially increase the interest of general readers, including the clinicians and experimental researchers:

-      Circulation. 1998 Jan 27;97(3):276-81. doi: 10.1161/01.cir.97.3.276

-      Apoptosis. 2006 Dec;11(12):2195-204. doi: 10.1007/s10495-006-0292-5

-      J Cell Mol Med. 2013 Aug;17(8):936-57. doi: 10.1111/jcmm.12074.

-      Front Pharmacol. 2022 Nov 14;13:964475. doi: 10.3389/fphar.2022.964475. eCollection 2022

-      Free Radic Biol Med. 2023 Feb 1;195:219-230. doi: 10.1016/j.freeradbiomed.2022.12.097.

The incorporation of the aforementioned publications and the suggested Figure in the revised version of the manuscript may substantially increase the interest of general readers, including the clinicians and experimental researchers.

  • We have cited these publications, we thank the reviewer, and they were very useful for revealing the topic of the review.
